# Defying Multi-model Forgetting: Orthogonal Gradient Learning to One-shot Neural Architecture Search

## Abstract

One-shot neural architecture search (NAS) trains an over-parameterized network (termed as supernet) that assembles all the architectures as its subnets by using weight sharing, and thereby reduces much computational budget. However, there is an issue of multi-model forgetting about supernet training in one-shot NAS that some weights of the previously well-trained architecture will be overwritten by that of the newly sampled architecture which has overlapped structures with the old one. To overcome the issue, we propose an orthogonal gradient learning (OGL) guided supernet training paradigm for one-shot NAS, where the novelty lies in the fact that the weights of the overlapped structures of current architecture are updated in the orthogonal direction to the gradient space of these overlapped structures of all previously trained architectures. Moreover, a new approach of calculating the projection is designed to effectively find the base vectors of the gradient space to acquire the orthogonal direction. We have theoretically and experimentally proved the effectiveness of the proposed paradigm in overcoming the multi-model forgetting. Besides, we apply the proposed paradigm to two one-shot NAS baselines, and experimental results have demonstrated that our approach is able to mitigate the multi-model forgetting and enhance the predictive ability of the supernet in one-shot NAS with remarkable efficiency on popular test datasets.

## 1 Introduction

Recent years, one-shot neural architecture search (one-shot NAS) has aroused massive interests and attentions in automatic architecture design, due to its remarkable efficiency in finding high-performance neural architectures under specific resource constraints Bender et al. (2018); Pham et al. (2018); Brock et al. (2017). The key of the one-shot NAS is weight sharing Guo et al. (2020a); Dong & Yang (2019a); Guo et al. (2020b); Chu et al. (2021), where the weights of all candidate architectures directly inherit from a supernet without training from scratch Hu et al. (2021); Chen et al. (2021); Yu et al. (2021). In this way, only the supernet needs to be trained during the architecture search, and the training time can be reduced from days to several hours, so that the search efficiency is greatly improved Yu et al. (2020); Dong & Yang (2019b).

Although the weight sharing can significantly enhance the computational efficiency, it may also introduce multi-model forgetting (defined by Benyahia et al. (2019)). During the training of a supernet, a number of architectures are sequentially sampled from the supernet and trained independently. Once the architectures have partially overlapped structures, the weights of these overlapped structures of the previously well-trained architecture will be overwritten by the weights of the newly sampled architectures. In this way, the performance of the previously well-trained architecture may be decreased.

The above phenomenon has also been verified by Zhang et al. (2020b;a) that the multi-model forgetting may ultimately lead to unreliable ranking of the validation accuracy due to the performance degradation in the training. An example to illustrate the above phenomenon is given in Fig. 1 that the observed validation accuracy of $ArcA$ is lower than that of baseline ($ArcA$) when $ArcB(n = 1)$ or ($n = 2$) is training. Besides, the more overlapped operations between $ArcA$ and $ArcB$ will trigger a larger degradation of the validation accuracy. This reason lies to the fact that the weights of

the overlapped operations of $ArcA$ are overwritten by the corresponding weights of $ArcB(n = 1)$ or $(n = 2)$. Therefore, how to defy the multi-model forgetting in one-shot NAS based on weight-sharing paradigm is essential.

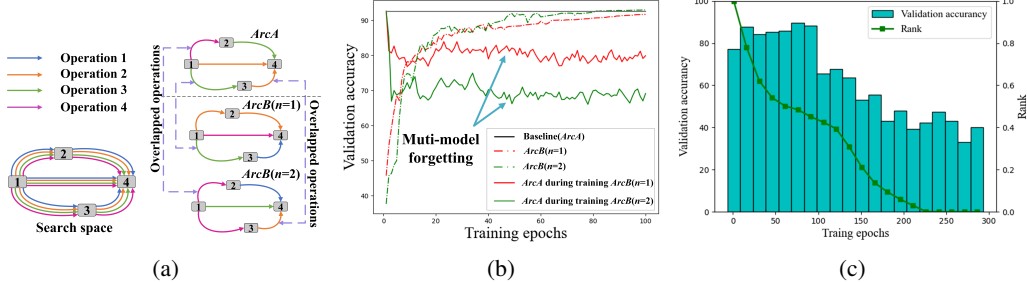

(a)  (b)  (c)

Figure 1: (a) $ArcA$ is firstly sampled from the search space, and then $ArcB(n = 1)$ or $ArcB(n = 2)$ is sampled in the next. (b) The curves of baseline ($ArcA$), $ArcB(n = 1)$ and $ArcB(n = 2)$ represent their validation accuracy during the training. The curve of $ArcA$ during training $ArcB(n = 1)$ or the training of $ArcB(n = 2)$ indicates the observed validation accuracy of $ArcA$ when training $ArcB(n = 1)$ or $(n = 2)$. (c) Validation accuracy of sampled architectures and average rank of the orthogonal projectors of overlapped structures with respect to the training epochs by the disturbance-immune (DI) Niu et al. (2021).

At present, the study of defying multi-model forgetting in one-shot NAS is in its infancy Benyahia et al. (2019). The typical way in handling this issue is to regularize the training of an architecture sampled from the supernet by adding constraints to the loss function Benyahia et al. (2019); Zhang et al. (2020b;a). These constraints designed to penalize the large changes of the weights of the over-lapped structures are constructed on a subset of an archive which preserves all previously trained architectures Zhang et al. (2020b). However, the above training process is subject to the size of subset, especially when it is filled up. Differently, recently proposed disturbance-immune (DI) Niu et al. (2021) introduces an idea to mitigate the multi-model forgetting issue. It introduces a projector to update the weights of the overlapped structures of current architecture in the orthogonal direction to the input space of the overlapped structures of all previously trained architectures. However, we discover that the DI has a potential issue of projector attenuation that the rank of the orthogonal projectors is approaching zero, and more overlapped cases will happen as the training process continues. Unfortunately, the projector attenuation will ultimately and severely degrade the efficiency of the weight learning as well as the prediction ability of the supernet.

In order to defy the multi-model forgetting of one-shot NAS and overcome the projector attenuation issue of DI, we develop an orthogonal gradient learning (OGL) paradigm for supernet training. During the training process, we firstly detect the structures of current architecture whether they are ever sampled or not. If not, then the weights of the architecture are updated with back-propagation algorithm (e.g., SGD) and a pre-constructed gradient space is updated by the gradient direction of the structure. If yes, then the weights of the overlapped structures of the current architecture are updated in the orthogonal direction to the gradient space of all previously trained architectures. Following the OGL guided training paradigm, the training of the current architecture will largely eliminate the influence to the performance of all previously trained architectures when they have overlapped structures, which has been theoretically and experimentally proved in this study.

To the best of our knowledge, it is the first attempt to use the orthogonal gradient to handle the multi-model forgetting problem in one-shot NAS and the main contributions include:

- An orthogonal gradient learning (OGL) guided supernet training paradigm is proposed to effectively defy the multi-model forgetting of one-shot NAS, which has been theoretically and experimentally proved that the OGL guided training paradigm enables the training of the current architecture to largely eliminate the impact to the performance of all previously trained architectures.

- An enhanced projection based on PCA is designed to find a set of base vectors to represent the gradient space of all previously trained architectures, which overcomes the projector

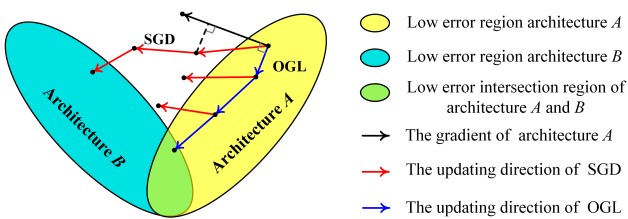

Figure 2: Difference between SGD and OGL in the training of the overlapped structures of architecture $B$ after the training of architecture $A$. The updating direction of the weights in SGD is towards the low error region of architecture $B$, and that in OGL is towards the intersection region where both architectures $A$ and $B$ have low test error.

attenuation issue and helps to determine the orthogonal direction without need to store the gradient vectors of all previously trained architectures.

- The proposed paradigm is integrated into two baselines, and a number of experimental results show that the OGL is able to mitigate the multi-model forgetting and improve the prediction ability of the baselines.

## 2 RELATED WORK

### 2.1 ONE-SHOT NAS AND ISSUE OF MULTI-MODEL FORGETTING

In the training of one-shot NAS, each candidate architecture $\alpha$ inherits weights from the supernet $\mathcal{W}_{\mathcal{A}}$ and directly evaluates it on the validation dataset without training. Afterwards, the best candidate architecture $\alpha^*$ and the weights of supernet $\mathcal{W}_{\mathcal{A}}(\alpha^*)$ can be found with the highest validation accuracy, as defined as

$$(\alpha^*, \mathcal{W}_{\mathcal{A}}(\alpha^*)) = \arg\min_{\alpha, \mathcal{W}} L_{train}(\mathcal{W}_{\mathcal{A}}(\alpha)), \tag{1}$$

where $L_{train}$ is the training loss function.

In the training of one-shot NAS, the issue of multi-model forgetting Benyahia et al. (2019) easily happens that the weights of the overlapped structures of the previously trained architecture are overwritten by the weights of the new architecture. As a consequence, the accuracy of previously trained architectures is seriously degraded after the training of a new architecture Yu et al. (2020); Li et al. (2020); Wang et al. (2020) and even lead to unreliable ranking of the validation accuracy Zhang et al. (2020b;a).

To handle multi-model forgetting, Benyahia et al. (2019) propose a weight plasticity loss (WPL), which regularizes the learning of the weights of overlapped structures according to the importance of the previously and firstly-trained architecture. However, Zhang et al. (2020b;a) have verified that considering only one previous architecture in each step seems not much effective in handling the multi-model forgetting issue so that they add constraints to the loss function to regularize the supernet training, where the constraints are referring to the loss functions of a subset of the previously trained architectures. Unfortunately, it is challenging to find an archive with appropriate size to represent all previously trained architectures during the training process. The recently proposed disturbance-immune (DI) Niu et al. (2021) shows a great potential to alleviate the multi-model forgetting, which updates the weights of each sampled operation in the orthogonal direction of input features through projection matrices. However, this approach suffers the issue of projector attenuation, i.e., the rank of the projectors will approach zero as the training process goes. In other words, the weights of the operation will be no more modified in the following training so that the prediction ability of the supernet will be degraded Zeng et al. (2019).

According to the above analysis, the key to handle the multi-model forgetting is to design an effective weight learning method and meanwhile avoid the projector attenuation.

## 3 PROPOSED METHOD

### 3.1 ORTHOGONAL GRADIENT LEARNING (OGL)

In this study, the cell structure of DARTS Saha et al. (2020) with 7 nodes is adopted to construct the standard supernet[1]. In other words, a supernet constitutes of a number of stacked cells. The cell structure includes two input nodes (nodes 0 and 1), four intermediate nodes (nodes 2, 3, 4 and 5), and one output node (node 6). The edge highlighted in dashed line indicates one of eight possible operations, i.e., $3 \times 3$ max pooling, $3 \times 3$ average pooling, $3 \times 3$ and $5 \times 5$ separable convolutions, $3 \times 3$ and $5 \times 5$ dilated separable convolutions, identity and zero. On the basis of the above cell structure, an architecture can be sampled from a supernet by selecting a subset of the dashed lines and determining the operation of each selected dashed line in the cells.

In order to handle the multi-model forgetting, we design an effective orthogonal gradient learning (OGL) for supernet training and meanwhile avoid the projector attenuation. The main idea is to design a gradient space to save the gradient directions of each operation of the previously trained architectures. Notably, the weight update of every operation is orthogonal to the gradient space. As a result, the impact from the weight update of the newly sampled architectures to the performance of the previously well-trained architectures will be largely eliminated. Consequently, the multi-model forgetting in one-shot NAS can be alleviated. Figure 2 shows an example to illustrate the above idea. In OGL, the updating direction of the weights in each update is determined by the direction of stochastic gradient descent (SGD). After a number of updates, we can see that the direction of OGL is towards the intersection region where both architectures $A$ and $B$ have low test error.

In SGD, the weights of architectures during the supernet training is obtained as follows:

$$w_{l,r}^{(i,j)}(k+1) \leftarrow w_{l,r}^{(i,j)}(k) - \eta \Delta w_{l,r}^{(i,j)}(k+1)^{BP}, \tag{2}$$

where $w_{l,r}^{(i,j)}(k)$ and $w_{l,r}^{(i,j)}(k+1)$ are the weight vectors of the $r$-th candidate operation $o_r^{(i,j)}$ between nodes $i$ and $j$ in cell $l$ for the $k$-th and $(k+1)$-th sampled architectures, respectively. $\eta$ is the learning rate. $\Delta w_{l,r}^{(i,j)}(k+1)^{BP}$, the weight direction of SGD calculated by back propagation (BP), is represented by the weight adjustment (or gradient) of $o_r^{(i,j)}$ in cell $l$ of architecture $k+1$.

In OGL, we modify the weight direction $\Delta w_{l,r}^{(i,j)}(k+1)^{BP}$ in Eq. 2 as follows:

$$\Delta w_{l,r}^{(i,j)}(k+1)^{BP} \leftarrow \Delta w_{l,r}^{(i,j)}(k+1)^{BP} - pro_{S_r^{(i,j)}}(\Delta w_{l,r}^{(i,j)}(k+1)^{BP}), \tag{3}$$

where $S_r^{(i,j)}$ is the gradient space of $o_r^{(i,j)}$. $pro_{S_r^{(i,j)}}(\Delta w_{l,r}^{(i,j)}(k+1)^{BP})$ represents the vector $\Delta w_{l,r}^{(i,j)}(k+1)^{BP}$ projected on $S_r^{(i,j)}$.

***Lemma* 1.** The OGL guided training paradigm enables the training of the current architecture to largely eliminate the impact to the performance of all previously trained architectures.

**Proof.** Define $w(k)$ as the weights of $k$-th architecture and $L(w)$ is the loss function. The loss of $k+1$-th architecture can be calculated according to Taylor formula:

$$L(w) = L(w(k)) + (w - w(k))L'(w(k)) + R_1(w), \tag{4}$$

where $L'(w)$ is the first derivative of $L(w)$, and $R_1(w) = o(w - w(k))$ is the remainder term of Eq. 4.

Note that $o(w - w(k))$ is approximately equal to zero when $w$ approaches $w(k)$. Then the Eq. 4 can be reformed as:

$$L(w) - L(w(k)) \approx (w - w(k))L'(w(k)). \tag{5}$$

where the left side of the Eq. 5 is due to the amount of change in the loss of the $k$-th architecture when training $k+1$-th architecture.

OGL guided training paradigm enables the modification of weights in the orthogonal direction of the corresponding gradient space. Every gradient space consists of the gradient direction

---

[1]Other structures of supernets can also be used in this work

(i.e., $L^{'}(w(k))$) of previous architectures. If the modification of operation weights $(w - w(k))$ of new architecture is updated in the orthogonal direction of its gradient space, we will have $\langle w - w(k), L^{'}(w) \rangle = 0$, and then $L(w) - L(w(k)) \to 0$, which means the performance of previously trained architectures is slightly impacted. The *Lemma* 1 is proven.

We further provide with the convergence analysis of the update of the weights in appendix **?**.

***Theorem* 1.** Given a $l$-smooth and convex loss function $L(w)$, $w^*$ and $w_0$ are the optimal and initial weights of $L(w)$, respectively. If we let the learning rate $\eta = 1/l$, then we have:

$$L(w_t) - L(w^*) \leq \frac{2l}{t} \|w_0 - w^*\|_F^2, \tag{6}$$

where $w_t$ is the weights of architecture after $t$-th training.

*Theorem* 1 demonstrates that our proposed method OGL has a convergence rate of $O(1/t)$ Liu et al. (2018b). The proof of *Theorem* is provided in the section 1.2 of the supplementary file.

## 3.2 DESIGN OF GRADIENT SPACE

In this section, we propose an orthogonal gradient learning algorithm that makes the update of the architecture parameters orthogonal to the gradient vectors, in order to preserve the gradients of the previously trained architectures. Specifically, we calculate the gradient of each operation of architecture $k$ after training $k$-th architecture and introduce the gradient vector to the update of the gradient space. Therefore, the term $pro_{S_r^{(i,j)}}(\Delta w_{l,r}^{(i,j)}(k+1)^{BP})$ in Eq. 3 can be obtained according to *Lemma* 2.

***Lemma* 2.** Given a gradient space $S_r^{(i,j)}$ consists of a number of gradient vectors, i.e., $S_r^{(i,j)} = \{g_1, g_2, ..., g_n\}$, the projection of $\Delta w_{l,r}^{(i,j)}(k+1)^{BP}$ on $S_r^{(i,j)}$ can be calculated by Eq. 6.

$$pro_{S_r^{(i,j)}}(\Delta w_{l,r}^{(i,j)}(k+1)^{BP}) = G(G^T G)^{-1} G^T \Delta w_{l,r}^{(i,j)}(k+1)^{BP}, \tag{7}$$

where $G = [g_1, g_2, ..., g_n]$, $g_i \in \mathbb{R}^{h \times 1}$, $i = 1, 2, ..., n$. $n$ and $h$ are the number of gradient vectors and the dimension of the gradient space $S_r^{(i,j)}$, respectively. The proof of *Lemma* 2 is provided in the section 1.1 of the supplementary file.

Notably, it is impractical to record all the gradient vectors during the training of the supernet, since the same operation is often repeatedly sampled. To address this issue, we describe $G$ in Eq. 6 with a small number of vectors since any vector in a space can be represented by a set of bases. Therefore, we perform PCA on the matrix $G \in \mathbb{R}^{h \times n}$ to get the set of representative bases $G_{dim} \in \mathbb{R}^{h \times d}$, where $d$ is the number of base vectors. After that, $G_{dim}$ is adopted in Eq. 6 to obtain the enhanced calculation of the projection as follows:

$$pro_{S_r^{(i,j)}}(\Delta w_{l,r}^{(i,j)}(k+1)^{BP}) = G_{dim}(G_{dim}^T G_{dim})^{-1} G_{dim}^T \Delta w_{l,r}^{(i,j)}(k+1)^{BP}. \tag{8}$$

Algorithm 1 outlines the training process of supernet with the proposed OGL.

## 3.3 RELATION WITH DI

In OGL, the weights of the overlapped structures of the current architecture are updated in the orthogonal direction to the gradient space of all previously trained architectures. By contrast, the disturbance-immune (DI) strategy proposed by Niu et al. (2021) is that the weights of an architecture are learnt in the orthogonal direction to the input space of the previously trained architectures. However, OGL and DI are fundamentally different: OGL updates the weights in the direction orthogonal to the gradient space, while DI is to learn the direction orthogonal to the input space. In DI, the recursive least square algorithm is used to calculate the projector, which may easily lead to the projector attenuation issue, i.e., the rank of orthogonal projectors will be quickly reduced to zero Zeng et al. (2019). As a consequence, the learning capacity of DI in defying the multi-modal forgetting issue is severly limited. An example is shown in Fig. 1c that the validation accuracy seriously decreases with the reduction of the rank value. By contrast, OGL is free from this projector attenuation issue by using Eq. 7 to calculate $pro_{S_r^{(i,j)}}(\Delta w_{l,r}^{(i,j)}(k+1)^{BP})$.

---

**Algorithm 1** OGL Supernet Training

---

**Input**: $D_{train}$: the training dataset, $D_{val}$: the validation dataset, $T$: NAS iteration
**Output**: The optimal architecture found by the proposed method OGL

1: Initialize the supernet weights $w_{l,r}^{(i,j)}(0)$ and archive $Arc = \emptyset$;
2: **for** $k = 0$ to $T - 1$ **do**
3:     Randomly sample an architecture from the supernet;
4:     Forward propagate all the input;
5:     Calculate $\Delta w_{l,r}^{(i,j)}(k)^{BP}$ through BP method;
6:     **if** non-overlapped operations **then**
7:         Update the weights using $\Delta w_{l,r}^{(i,j)}(k)^{BP}$.
8:     **else**
9:         Update the weights using Eq. 3 and Eq. 6.
10:     **end if**
11:     Perform PCA on each operation and update the gradient space with the gradient vectors.
12:     Update $Arc$ by new gradient spaces.
13: **end for**
14: Obtain the optimal architecture by Eq. 1.

---

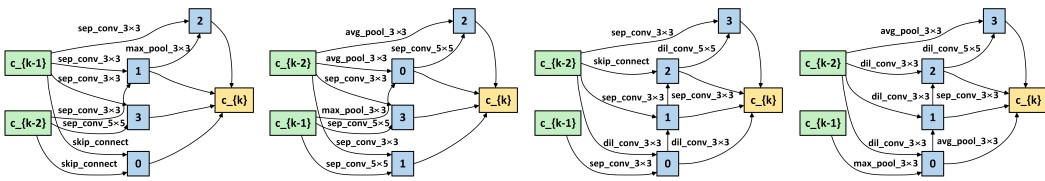

(a) Normal cell searched by RandomNAS-OGL
(b) Reduction cell searched by RandomNAS-OGL
(c) Normal cell searched by GDAS-OGL
(d) Reduction cell searched by GDAS-OGL

Figure 3: The best cells discovered on CIFAR-10.

## 4 EXPERIMENT

### 4.1 ONE-SHOT NAS WITH OGL

In this study, we apply OGL to two popular single-path one-shot NAS baselines, including the RandomNAS Li & Talwalkar (2020) and GDAS Dong & Yang (2019b), where RandomNAS and GDAS are random- and gradient-based sampling NAS methods, respectively. For convenience, our proposed methods based on the baselines are denoted by RandomNAS-OGL and GDAS-OGL. In the experiments, we compare our methods with 13 state-of-the-art one-shot NAS competitors on CIFAR-10, CIFAR-100 and ImageNet, where the experimental settings (e.g., the search space and hyperparameters) follow Dong & Yang (2019b); Li & Talwalkar (2020) for fair comparison.

### 4.2 RESULTS ON CIFAR-10 AND CIFAR-100

In this section, the experimental results on CIFAR-10 and CIFAR-100 are presented. Note that we use 8 cells with 16 initial channels and 64 batch size to construct the supernet. The final architecture is composed of 20 cells, which is trained with 96 batch size. The whole architecture search, including operation search and topology search, consumes 41.2 hours (1.7 GPU-Days) with 240 epochs on an NVIDIA Tesla V100 GPU. Table 1 shows the comparison results.

- Compared with the baselines (RandomNAS and GDAS), RandomNAS-OGL and GDAS-OGL show better performance in terms of the test loss and model size. Specifically, the test loss values on CIFAR-10 of RandomNAS and GDAS decrease from 2.85% and 2.93% to 2.63% and 2.83%, respectively. Also, the test error on CIFAR-100 of RandomNAS and GDAS decrease from 17.63% and 18.38% to 17.54% and 17.75%, respectively. Their corresponding model parameters decrease from 4.3M and 3.4M to 3.52M and 3.14M, respectively. Accordingly, OGL can increase the test accuracy and meanwhile decrease the

| Method | Test Error(%) | | Paras. | FLOPs | Search Cost | Memory | Supernet |
|---|---|---|---|---|---|---|---|
| | CIFAR-10 | CIFAR-100 | (M) | (M) | (GPU Days) | Consumption | Optimization |
| ENASPham et al. (2018) | 3.54 | 19.43† | 4.6 | - | 0.45 | Single Path | RL |
| NAO-WSLuo et al. (2018) | 3.53 | - | 2.5 | - | - | Single Path | Gradient |
| SNASXie et al. (2018) | 2.85±0.02 | 20.09* | 2.8 | 422 | 1.5 | Whole Supernet | Gradient |
| PARSECCasale et al. (2019) | 2.86±0.06 | - | 3.6 | 485 | 0.6 | Single Path | Gradient |
| BayesNASZhou et al. (2019) | 2.81±0.04 | - | 3.4 | - | 0.2 | Whole Supernet | Gradient |
| RENASChen et al. (2019b) | 2.88±0.02 | - | 3.5 | - | 6 | - | RL&EA |
| MdeNASZheng et al. (2019) | 2.87 | 17.61* | 3.78 | 599 | 0.16 | Single Path | MDL |
| DSO-NASZhang et al. (2020c) | 2.87±0.07 | - | 3.0 | - | 1 | Whole Supernet | Gradient |
| Random BaselineLiu et al. (2018b) | 3.29±0.15 | - | 3.2 | - | 4 | - | Random |
| DARTS(1st)Liu et al. (2018b) | 2.94 | - | 2.9 | 501 | 1.5 | Whole Supernet | Gradient |
| DARTS(2nd)Liu et al. (2018b) | 2.76±0.09 | 17.57† | 3.4 | 528 | 4 | Whole Supernet | Gradient |
| WPLBenyahia et al. (2019) | 3.81 | - | - | - | - | Single Path | RL |
| DI-RandomNAS + cutoutNiu et al. (2021) | 2.87±0.04 | 17.71‡ | 3.6 | - | 1.5 | Single Path | Random |
| RandomNAS-NSASZhang et al. (2020a) | 2.64 | 17.56 | 3.08 | 489 | 0.7 | Single path | Random |
| GDAS-NSASZhang et al. (2020a) | 2.73 | 18.02 | 3.54 | 528 | 0.4 | Single path | Gradient |
| GDASDong & Yang (2019b) | 2.93 | 18.38 | 3.4 | 519 | 0.21 | Single Path | Gradient |
| GDAS-OGL | 2.83 | 17.75 | 3.14 | 528 | 0.3 | Single Path | Gradient |
| RandomNASLi & Talwalkar (2020) | 2.85±0.08 | 17.63* | 4.3 | 612 | 2.7 | Single Path | Random |
| RandomNAS-OGL | **2.63±0.02** | **17.54** | 3.52 | 503 | 0.5 | Single Path | Random |

Table 1: Comparision results in terms of test error on CIFAR-10 and CIFAR-100. "*" indicates that the results are reported in the Zhang et al. (2020a). "†" indicates that the results are reported in the Dong & Yang (2019b). "‡" indicates that the results are reported by ourselves. "-" indicates that these methods are not reproducd in the experiment. All models are trained with 600 epochs except RandomNAS-OGL, which is trained with 1000 epochs to get the optimal results.

model parameters. These results initially show the effectiveness of OGL in overcoming the multi-model forgetting.

- Compared with the popular multi-model forgetting methods, i.e., WPL Benyahia et al. (2019), NSAS Zhang et al. (2020a) and DI Niu et al. (2021), our methods achieve better performance, i.e., getting lower test error with fewer parameters. Specifically, RandomNAS-OGL and GDAS-OGL outperforms WPL with a test error improvement of 1.18% and 0.98% on CIFAR-10, respectively. Also, RandomNAS-OGL performs better than RandomNAS-NSAS with 0.01% test error improvement on CIFAR-10 and 0.02% test error improvement on CIFAR-100. Furthermore, GDAS-OGL performs better than GDAS-NSAS on CIFAR-100 with smaller test error (0.27% improvement) and less parameters (0.4M improvement). In addition, RandomNAS-OGL outperforms DI-RandmNAS with a test error improvement of 0.24% and 0.17% on CIFAR-10 and cifar-100, respectively.

- Compared with other NAS methods, RandomNAS-OGL and GDAS-OGL exhibit competitive performance. Especially, RandomNAS-OGL gets 2.63% test error on CIFAR-10 and 17.54% test error on CIFAR-100, and GDAS-OGL achieves 2.83% on CIFAR-10 and 17.75% on CIFAR-100, which are better than most of compared methods. The results indicate that OGL is capable of enhancing the supernet prediction.

- Fig. 3 presents the visualization results of the best cells found by RandomNAS-OGL for CNN models on CIFAR-10 and CIFAR-100.

## 4.3  RESULTS ON IMAGENET

This section is to evaluate the transferability of OGL, where the architecture discovered from CIFAR-10 will be tested on ImageNet dataset. Here, RandomNAS is employed as the baseline. In this experiment, we train the architecture in 250 epochs with 52 initial channels and 128 batch size. Table 2 shows the comparison results between our method and other NAS methods with or without weight sharing. The results show that the test error of RandomNAS-OGL is 25.8%, which outperforms the baseline (RandomNAS) with 1.3%. In comparision with other methods, RandomNAS-OGL is on par with or even better than them. Specifically, our proposed method RandomNAS-OGL outperforms RandomNAS-NSAS with 0.3% improvement and outperforms DI-RandomNAS with 0.6% improvement. In average, RandomNAS-OGL performs better than most of NAS methods by 0.3%-1.2%. These results demonstrate OGL has a great transferabilityof the searched convolutional

cells, and also show that OGL plays a positive role in improving the prediction ability of the one-shot NAS method (RandomNAS).

| Method | Test Error (%) | Param.(M) | FLOPs(M) |
|---|---|---|---|
| Inception-v1Szegedy et al. (2015) | 30.2 | 6.6 | 1448 |
| MobileNetHoward et al. (2017) | 29.4 | 4.2 | 569 |
| ShuffleNet 2 ×Zhang et al. (2018) | 26.4 | 5 | 524 |
| NASNet-AZoph et al. (2018) | 26.0 | 5.3 | 564 |
| PNASLiu et al. (2018a) | 25.8 | 5.1 | 588 |
| SNASXie et al. (2018) | 27.3 | 4.3 | 522 |
| PARSECCasale et al. (2019) | 26.3 | 5.5 | - |
| BayesNASZhou et al. (2019) | 26.5 | 3.9 | - |
| MdeNASZheng et al. (2019) | 26.8 | 6.1 | 595 |
| DSO-NASZhang et al. (2020c) | 26.2 | 4.7 | 571 |
| PDARTSChen et al. (2019a) | 25.9* | 4.9 | 557 |
| DARTS(2nd)Liu et al. (2018b) | 26.7 | 4.7 | 574 |
| DI-RandomNAS + cutoutNiu et al. (2021) | 26.4‡ | 5.1 | 587 |
| RandomNAS-NSASZhang et al. (2020a) | 26.1 | 5.2 | 581 |
| GDAS-NSASZhang et al. (2020a) | 26.7 | 5.1 | 564 |
| RandomNASLi & Talwalkar (2020) | 27.1 | 5.4 | 595 |
| RandomNAS-OGL | **25.8** | 5.9 | 589 |

Table 2: Comparision results on Imagenet. The first block contains the NAS methods without weight sharing. The second block contains one-shot NAS methods. "*" indicates that the results are reported in the Li & Talwalkar (2020). "‡" indicates that the results are reported by ourselves.

## 4.4 MULTI-MODEL FORGETTING IN ONE-SHOT NAS

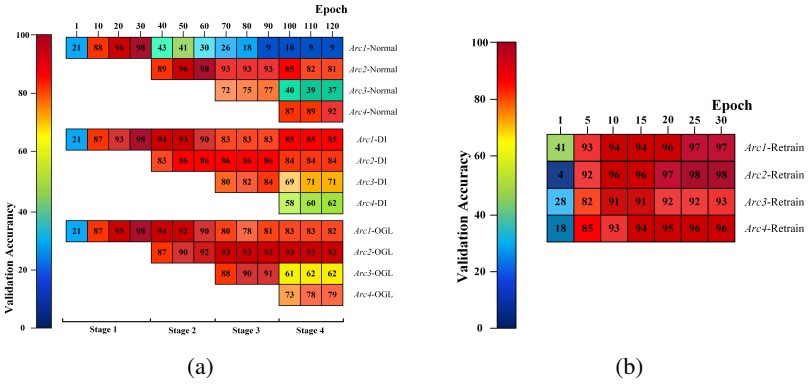

Figure 4: (a) Validation accuracy comparison of normal, DI and OGL supernet training. Four architectures $Arc1$, $Arc2$, $Arc3$, and $Arc4$ are sampled and trained sequentially from stage 1 to stage 4. At each stage, only one architecture is trained for updating the supernet, while each block is the validation accuracy of corresponding architecture tested on MNIST. (b) The performance comparison of the four architectures $Arc1$, $Arc2$, $Arc3$, and $Arc4$ during the retraining.

In order to intuitively observe the effectiveness of our methods in relieving multi-model forgetting in one-shot NAS, we test the validation accuracy of previously-trained architectures during a new architecture is training. As shown in Fig. 4 (a), given four architectures ($Arc1$, $Arc2$, $Arc3$, and $Arc4$) which have overlapped structures, we will track their validation accuracy when they are trained in a sequential way by using OGL, DI or normal supernet training. Specifically, $Arc1$ is firstly trained to update the supernet at stage 1 (1-30 epochs); At stage 2 (31-60 epochs), only $Arc2$ is trained for the supernet, while the validation accuracy of $Arc1$ through inheriting supernet weights is tracked; Similarly, at stage 3 (61-90 epochs) or stage 4 (91-120 epochs), only $Arc3$ or $Arc4$ is trained for updating the supernet, while the validation accuracy of other architectures is recorded.

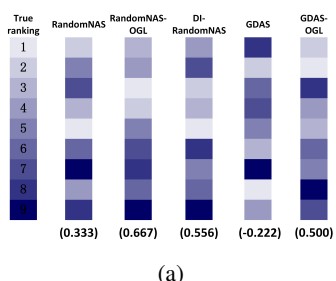 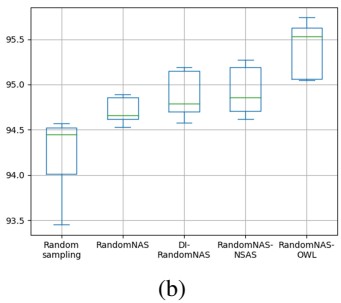

| (a) | (b) |

Figure 5: (a) The final Kendall Tau values and architecture ranking of different methods with or without OGL. (b) The mean validation accuracy for the architectures found through different methods.

Fig. 4 (a) shows the validation accuracy changes of the four architectures on MNIST. We observe that the validation accuracy results of $Arc1$-Normal, $Arc2$-Normal, $Arc3$-Normal drop drastically at stage 2, 3, and 4, respectively. It indicates that the normal supernet training is easily impacted by the multi-model forgetting issue since these architectures have overlapped structures. In contrast, the validation accuracy of the architectures with DI and OGL remains stable to much extent when a new architecture is training. These results show that DI and OGL both are effective to prevent multi-model forgetting. However, DI may lead to unreliable ranking of architectures due to the projector attenuation issue. As shown in Fig. 4 (b), the true validation accuracy rank of the four architectures after retraining from the scratch is $Arc2$, $Arc1$, $Arc4$, and $Arc3$. In Fig. 4 (a), we can see that the final accuracy rank of these architectures trained by normal supernet training is $Arc4$, $Arc2$, $Arc3$ and $Arc1$, which is totally different from the true ranking due to the multi-model forgetting. And the result of DI ($Arc1$, $Arc2$, $Arc3$, and $Arc4$) is different from the true ranking due to the projector attenuation issue. However, the result of OGL can lead to reliable rank by defying the multi-model forgetting and projector attenuation, which is the key to assure the supernet predictive ability.

### 4.5 SUPERNET PREDICTIVE ABILITY COMPARISON

The supernet predictive ability can be measured by the correlation between the architecture rankings obtained by weight sharing and retraining. This correlation is commonly evaluated by Kendall Tau $\tau$ metric Zheng et al. (2019); Kendall (1945) where $\tau$ close to 1.0 means high correlation and strong predictive ability. Therefore, we compare the architecture rankings of the methods with or without OGL. Firstly, we sample 9 promising architectures (3 from RandomNAS, 3 from RandomNAS-OGL, and 3 randomly sampled from previous experiment) and retrained them from scratch. Similar experiments are also conducted on GDAS and GDAS-OGL.

Fig. 5 (a) presents the Kendall Tau $\tau$ metric of the architecture rankings based on normal method, OGL, DI and retraining (true ranking). These results show the difference in rankings between the retraining and the proposed method OGL. Specifically, Fig. 5 (a) presents the final Kendall Tau values $\tau$ of RandomNAS (0.333), RandomNAS-OGL (0.667), RandomNAS-DI (0.556), GDAS (-0.222) and GDAS-OGL (0.500). Note that the closer $\tau$ is to 1, the more perfect the consistency of the two sequences (the more similar of the two sequences). We can find that the architecture rankings of the methods with OGL is far more similar to the true ranking than that of the methods without OGL. And the result of RandomNAS-OGL is better than DI-RandomNAS. In other words, it indicates that the supernet obtained through OGL has a stronger predictive ability. Fig. 5 (b) depicts the mean validation accuracy of sampled architectures through different methods. The results show that the architectures of RandomNAS-OGL have better validation accuracy than that of other methods. Hence, we can conclude that OGL is able to find the architectures with high quality.

### 5 CONCLUSION AND FUTURE WORK

The goal of this work is to train a supernet in an effective way to overcome the multi-model forgetting in one-shot neural NAS. To this end, we proposed an orthogonal weight learning method to update the weights of current architecture in the direction orthogonal to the constructed gradient

space. In this way, the update of new weights will not impact the performance of previously-trained architectures and free from projector attenuation issue. A series of experiments have been conducted and the results have theoretically and experimentally demonstrated the effectiveness of OGL in improving the ability of prediction and defying the multi-model forgetting in one-shot NAS. In the future work, more effective way to store all gradient vectors of all architectures can be explored to indirectly improve the supernet predictive ability.

In future work, we will focus on the storage and computational issues. From all the experiments above, OGL shows a slightly higher FLOPs but relatively lower search costs (GPU days) in comparison with other methods. OGL aims to deal with multi-model forgetting issue and have to calculate the gradient vector of each operation and update the corresponding gradient space. Hence, the improvement of the accuracy (Test error) achieved by OGL is on the sacrifice of the complexity of the proposed method, leading to a higher FLOPs. Notably, the reasons for fewer GPU days of OGL is because the improvement of the accuracy can speed up the model convergence rate. From the results, OGL is relatively efficient in supernet training. For the storage issue, although PCA has reduced the dimension of the gradient space, OGL still needs to store a set of base vectors. Hence, we will develop more efficent calculation of orthogonal direction with fewer storage, with the consideration of using iterative methods instead of directly storing them.

# A  APPENDIX

## A.1  SUPPLEMENTARY EXPERIMENTS ON CIFAR-10 AND CIFAR-100

| Method | Test Error(%) | | Paras. | FLOPs | Search Cost | Memory | Supernet |
|---|---|---|---|---|---|---|---|
| | CIFAR-10 | CIFAR-100 | (M) | (M) | (GPU Days) | Consumption | Optimization |
| MetaQNN Baker et al. (2016) | 6.92 | 27.14 | 11.2 | - | >80 | Single Path | RL |
| NASNet-A Zoph et al. (2018) | 3.41 | 19.7 | 3.3 | 564 | 2000 | Single Path | RL |
| NASNet-A+cutout Zoph et al. (2018) | 2.65 | 17.81 | 3.3 | 564 | 2000 | Single Path | RL |
| SMASH Brock et al. (2017) | 4.03 | 20.6 | 16 | - | - | Single Path | Random |
| GDAS + cutout Dong & Yang (2019b) | 2.93 | 18.38 | 3.4 | - | 0.84 | Single Path | Gradient |
| ENASPham et al. (2018) | 3.54 | 19.43† | 4.6 | - | 0.45 | Single Path | RL |
| ENAS + cutoutPham et al. (2018) | 2.89 | 18.91† | 4.6 | - | 0.5 | Single Path | RL |
| SNASXie et al. (2018) | 2.85±0.02 | 20.09* | 2.8 | 422 | 1.5 | Whole Supernet | Gradient |
| MdeNASZheng et al. (2019) | 2.87 | 17.61* | 3.78 | 599 | 0.16 | Single Path | MDL |
| DARTS(2nd)Liu et al. (2018b) | 2.76±0.09 | 17.57† | 3.4 | 528 | 4 | Whole Supernet | Gradient |
| WPLBenyahia et al. (2019) | 3.81 | - | - | - | - | Single Path | RL |
| DI-RandomNAS + cutoutNiu et al. (2021) | 2.87±0.04 | 17.71‡ | 3.6 | - | 1.5 | Single Path | Random |
| RandomNAS-NSASZhang et al. (2020a) | 2.64 | 17.56 | 3.08 | 489 | 0.7 | Single path | Random |
| GDAS-NSASZhang et al. (2020a) | 2.73 | 18.02 | 3.54 | 528 | 0.4 | Single path | Gradient |
| GDASDong & Yang (2019b) | 2.93 | 18.38 | 3.4 | 519 | 0.21 | Single Path | Gradient |
| GDAS-OGL | 2.83 | 17.75 | 3.14 | 528 | 0.3 | Single Path | Gradient |
| RandomNASLi & Talwalkar (2020) | 2.85±0.08 | 17.63* | 4.3 | 612 | 2.7 | Single Path | Random |
| RandomNAS-OGL | **2.63±0.02** | **17.54** | 3.52 | 503 | 0.5 | Single Path | Random |

Table 3: Comparision results in terms of test error on CIFAR-10 and CIFAR-100. "*" indicates that the results are reported in the Zhang et al. (2020a). "†" indicates that the results are reported in the Dong & Yang (2019b). "‡" indicates that the results are reported by ourselves. "-" indicates that these methods are not reproducd in the experiment. All models are trained with 600 epochs except RandomNAS-OGL, which is trained with 1000 epochs to get the optimal results.

In order to evaluate the performance of the proposed OGL in handling the multi-model forgetting issue, we compared OGL with WPL, DI-RandomNAS, RandomNAS-NSAS and GDAS-NSAS, which are the existing methods in solving the multi-model forgetting issue. And the experimental results demonstrate that our OGL performs better in solving the multi-model forgetting.

In addition, we also compared OGL with ten state-of-the-art one-shot NAS methods without designing techniques to handle the multi-model forgetting issues. In Table 3, we have also provided extra six state-of-the-art one-shot NAS methods to get a comprehensive comparison. Specifically, RandomNAS-OGL gets 2.63% test error on CIFAR-10 and 17.54% test error on CIFAR-100, and GDAS-OGL achieves 2.83% on CIFAR-10 and 17.75% on CIFAR-100, which are better than most

of compared methods. The results show that the methods especially OGL with the techniques show better performance than other NAS methods without the techniques.

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
