# OVERCOMING MUTI-MODEL FORGETTING IN ONE-SHOT NEURAL ARCHITECTURE SEARCH VIA ORTHOGONAL GRADIENT LEARNING

## 1 SUPPLEMENTARY FILE.

### 1.1 A PROOF OF *Lemma* 2

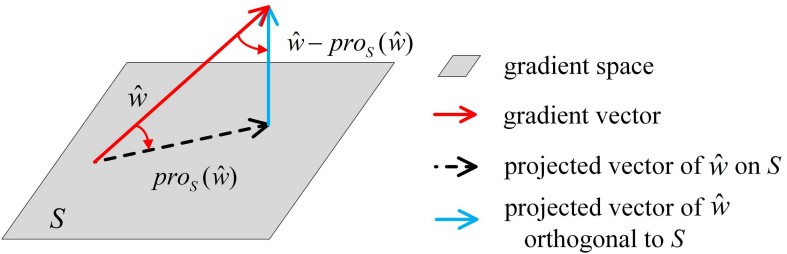

Figure 1: An illustration of the relationship between the input space $S$ and the orthogonal projector $P$, where $f = b - c$.

**Lemma 2.** Given a gradient space $S_r^{(i,j)}$ consists of a number of gradient vectors, i.e., $S_r^{(i,j)} = \{g_1, g_2, ..., g_n\}$, the projection of $\Delta w_{l,r}^{(i,j)}(k+1)^{BP}$ on $S_r^{(i,j)}$ can be calculated by Eq. 1.

$$pro_{S_r^{(i,j)}}(\Delta w_{l,r}^{(i,j)}(k+1)^{BP}) = $$
$$G(G^T G)^{-1} G^T \Delta w_{l,r}^{(i,j)}(k+1)^{BP}, \tag{1}$$

where $G = [g_1, g_2, ..., g_n]$, $g_i \in \mathbb{R}^{h \times 1}$, $i = 1, 2, ..., n$. $n$ and $h$ are the number of gradient vectors and the dimension of the gradient space $S_r^{(i,j)}$, respectively.

**Proof.** We use $\hat{w}$ to represent $\Delta w_{l,r}^{(i,j)}(k+1)^{BP}$ and use $S$ to represent the gradient space $S_r^{(i,j)}$ for simplicity. We take Figure 1 to illustrate the relationship between $\hat{w}$ and $S$. In the figure, $\hat{w}$ is a gradient vector, while $pro_S(\hat{w})$ is the gradient $\hat{w}$ projected on $S$.

*Step 1*: we express $pro_S(\hat{w})$ by a linear combination of the gradient vectors from $S$:

$$pro_S(\hat{w}) = x_1 g_1 + x_2 g_2 + ... + x_n g_n = GX, \tag{2}$$

where $G$ is a matrix of gradient vectors from $S$, i.e., $G = [g_1, g_2, ..., g_n]^T$. $X$ is a vector of constants, denoted by $X = [x_1, x_2, ..., x_n]^T$.

*Step 2*: In order to get $X$, we find that $\hat{w} - pro_S(\hat{w})$ is orthogonal to $S$. In other words, $\hat{w} - pro_S(\hat{w})$ is orthogonal to any input vector from $S$. Namely, the inner product of $\hat{w} - pro_S(\hat{w})$ and $g_i$ is zero, where $i = 1, \ldots, n$.

$$\begin{cases} <g_1, \hat{w} - pro_S(\hat{w})> = g_1^T \cdot (\hat{w} - GX) = 0 \\ ... \\ <g_n, \hat{w} - pro_S(\hat{w})> = g_n^T \cdot (\hat{w} - GX) = 0 \end{cases} \tag{3}$$

And we can reform the Eq. 3 by matrix calculations as follows:

$$G^T(\hat{w} - GX) = 0. \tag{4}$$

*Step 3*: On the basis of *Step 2*, we can get the vector $X$ as follows:

$$X = (G^T G)^{-1} G^T \hat{w}. \tag{5}$$

*Step 4*: We integrate Eq. 5 into Eq. 2 to get $pro_S(\hat{w})$ using

$$pro_S(\hat{w}) = GX = G(G^T G)^{-1} G^T \hat{w}. \tag{6}$$

Thus the *Lemma* 2 is proven.

## 1.2 CONVERGENCE GUARANTEE OF OGL

***Theorem*** **1.** Given a $l$-smooth and convex loss function $L(w)$, $w^*$ and $w_0$ are the optimal and initial weights of $L(w)$, respectively. If we let $\eta = 1/l$, then we have:

$$L(w_t) - L(w^*) \leq \frac{2l}{t} \|w_0 - w^*\|_F^2, \tag{7}$$

where $w_t$ is the weights after $t$-th training.

Theorem 1 demonstrates that our proposed method OGL has a convergence rate of $O(1/t)$ Niu et al. (2021).

**Proof.** Based on projected gradient descent (PGD) Nesterov (2003), the update of $w$ can be represented as follows:

$$q(w_t) = \arg\min_{w \in Q} (L(w_t) + \langle L'(w_t), w - w_t \rangle + \frac{\beta}{2} \|w_t - w\|_F^2) \tag{8}$$

$$w_{t+1} = w_t - \eta\beta(w_t - q(w_t)), \tag{9}$$

where $\eta$ and $\beta$ are two hyperparameters.

Let $Q$ is a closed convex set, $w^+ \in Q$ and $\beta \geq l$. We denote $Q_w = q(w^+)$ and $g_Q = g_Q(w^+) = \beta(w^+ - q(w^+))$, then we let:

$$\phi(w) = L(w^+) + \langle L'(w^+), w - w^+ \rangle + \frac{\beta}{2} \|w - w^+\|_F^2. \tag{10}$$

Based on Eq. 10, we have $\phi'(w) = L'(w^+) + \beta(w - w^+)$, Then we have:

$$\phi'(Q_w) = L'(w^+) + \beta(Q_W - w^+) = L'(w^+) - g_Q. \tag{11}$$

and

$$\langle L'(w^+) - g_Q, w - Q_w \rangle = \langle L'(w^+), w - Q_w \rangle - \langle g_Q, w - Q_w \rangle = \langle \phi'(Q_w), w - Q_w \rangle \geq 0. \tag{12}$$

Based on Eq. 10, Eq. 11 and Eq. 12 and the property of convex function, we have:

$$
\begin{aligned}
L(w) &\geq L(w^+) + \langle L^{'}(w^+), w - w^+ \rangle \\
&= L(w^+) + \langle L^{'}(w^+), w - Q_w \rangle + \langle L^{'}(w^+), Q_w - w^+ \rangle \\
&\geq L(w^+) + \langle L^{'}(w^+), Q_w - w^+ \rangle + \langle g_Q, w - Q_w \rangle \\
&= \phi(Q_w) - \frac{\beta}{2} \left\| Q_w - w^+ \right\|_F^2 + \langle g_Q, w - Q_w \rangle \\
&= \phi(Q_w) - \frac{1}{2\beta} \left\| g_Q \right\|_F^2 + \langle g_Q, w - Q_w \rangle \\
&= \phi(Q_w) - \frac{1}{2\beta} \left\| g_Q \right\|_F^2 + \langle g_Q, w - w^+ \rangle + \langle g_Q, \frac{1}{\beta} g_Q \rangle \\
&= \phi(Q_w) + \frac{1}{2\beta} \left\| g_Q \right\|_F^2 + \langle g_Q, w - w^+ \rangle.
\end{aligned}
\tag{13}
$$

And $\phi(Q_w) \geq L(Q_w)$ since $\beta \geq l$, Eq. 13 can be formulated as:

$$
L(w) \geq \phi(Q_w) + \frac{1}{2\beta} \left\| g_Q \right\|_F^2 + \langle g_Q, w - w^+ \rangle \geq L(Q_w) + \frac{1}{2\beta} \left\| g_Q \right\|_F^2 + \langle g_Q, w - w^+ \rangle. \tag{14}
$$

Based on Eq. 14, we let $\beta = l$, $w = w^+ = w_t$ and $L(q(w_t)) \geq L(q(w_{t+1}))$, then we have $\langle g_Q, w - w^+ \rangle = 0$ and:

$$
L(w_t) \geq L(w_{t+1}) + \frac{1}{2l} \left\| g_Q(w_t) \right\|_F^2, \tag{15}
$$

where $g_Q(w_t) = \beta(w_t - q(w_t))$.

Also, based on Eq. 14, we let $\beta = l$, $w = w^*$, $w^+ = w_t$ and $L(q(w_t)) \geq L(w^*)$, then we have:

$$
L(w^*) \geq L(w^*) + \frac{1}{2l} \left\| g_Q(w_t) \right\|_F^2 - \langle g_Q(w_t), w^* - w^t \rangle. \tag{16}
$$

We denote $r_t = \left\| w_t - w^* \right\|_F$ and $g_{Q,t} = g_Q(w_t)$, then based on Eq.16, we have:

$$
\begin{aligned}
r_{t+1}^2 &= \left\| w_{t+1} - w^* \right\|_F^2 \\
&= \left\| w_t - w^* - \eta g_{Q,t} \right\|_F^2 \\
&= r_t^2 - 2\eta \langle g_{Q,t}, w_t - w^* \rangle + \eta^2 \left\| g_{Q,t} \right\|_F^2 \\
&\leq r_t^2 - \frac{\eta}{\beta} \left\| g_{Q,t} \right\|_F^2 + \eta^2 \left\| g_{Q,t} \right\|_F^2 \\
&= r_t^2 + \eta(\eta - \frac{1}{\beta}) \left\| g_{Q,t} \right\|_F^2.
\end{aligned}
\tag{17}
$$

And based on Eq. 17 and if $\eta \leq 1/\beta$, then we have:

$$
r_{t+1}^2 \leq r_t^2 \leq r_{t-1} \leq \ldots \leq r_0^2. \tag{18}
$$

We denote $\Delta_t = L(w_t) - L(w^*)$ and based on Eq. 18, then we have:

$$
\Delta_t \leq \langle g_{Q,t}, w_t - w^* \rangle \leq r_0 \left\| g_{Q,t} \right\|_F. \tag{19}
$$

Based on Eq. 15 and Eq. 19, we have:

$$
\begin{aligned}
\Delta_{t+1} &= L(w_{t+1}) - L(w^*) \\
&\le L(w_t) - L(w^*) - \frac{1}{2l} \left\| g_{Q,t} \right\|_F^2 \\
&= \Delta_t - \frac{1}{2l} \left\| g_{Q,t} \right\|_F^2 \\
&\le \Delta_t - \frac{1}{2l} \frac{\Delta_t^2}{r_0^2}.
\end{aligned} \tag{20}
$$

Based on Eq. 20, we have:

$$
\frac{1}{\Delta_t} \le \frac{1}{\Delta_{t+1}} - \frac{1}{2l} \frac{1}{r_0^2} \frac{\Delta_t}{\Delta_{t+1}}. \tag{21}
$$

Based on Eq.21 combined with $\Delta_{t+1} \le \Delta_t$, we have:

$$
\frac{1}{\Delta_{t+1}} \ge \frac{1}{\Delta_t} + \frac{1}{2l} \frac{1}{r_0^2} \frac{\Delta_t}{\Delta_{t+1}} \ge \frac{1}{\Delta_t} + \frac{1}{2l} \frac{1}{r_0^2} \quad \ge \ldots \ge \frac{1}{\Delta_0} + \frac{1}{2l} \frac{1}{r_0^2}. \tag{22}
$$

Then we have:

$$
\begin{aligned}
\Delta_t &= L(w_t) - L(w^*) \\
&\le \frac{1}{\frac{1}{\Delta_0} + \frac{1}{2l} \frac{t}{r_0^2}} \\
&= \frac{1}{\frac{1}{L(w_0) - L(w^*)} + \frac{1}{2l} \frac{t}{\left\| w_0 - w^* \right\|_F^2}} \\
&= \frac{2l(L(w_0) - L(w^*)) \left\| w_0 - w^* \right\|_F^2}{2l \left\| w_0 - w^* \right\|_F^2 + t(L(w_0) - L(w^*))}.
\end{aligned} \tag{23}
$$

Based on the property of L-smooth function, we have:

$$
L(w_0) \le L(w^*) + \langle L^{'}(w^*), w_0 - w^* \rangle + \frac{l}{2} \left\| w_0 - w^* \right\|_F^2. \tag{24}
$$

We have $L(w_0) \ge L(w^*)$, combine with Eq. 23 and Eq. 24, we have:

$$
L(w_t) - L(w^*) \le \frac{2l \left\| w_0 - w^* \right\|_F^2}{2l \frac{\left\| w_0 - w^* \right\|_F^2}{L(w_0) - L(w^*)} + t} \le \frac{2l}{t} \left\| w_0 - w^* \right\|_F^2. \tag{25}
$$

Thus the *Theorem* 1 is proven.