# OpenReview forum: "Defying Multi-model Forgetting: Orthogonal Gradient Learning to One-shot Neural Architecture Search"
_ICLR.cc/2024/Conference — Submitted to ICLR 2024_

### Official Review · Reviewer_9XeX · 2023-10-28

**Soundness:** 3 good
**Presentation:** 3 good
**Contribution:** 3 good
**Rating:** 5
**Confidence:** 3

**Summary:**

In this paper, the main objective of the research is to train a supernet effectively to overcome the problem of multi-model forgetting in one-shot Neural Architecture Search (NAS). To address this issue, the authors propose a method called Orthogonal Gradient Learning (OGL) to update the weights of the current architecture in a way that they become orthogonal to the constructed gradient space. A series of experiments are conducted in this paper on multiple datasets to evaluate the effectiveness of OGL in addressing the multi-model forgetting problem.

**Strengths:**

（1）The logic of this paper is clear and it is easy to read.
（2）OGL offers a fresh perspective for one-shot NAS, especially in addressing the multi-model forgetting issue. Compared to existing suboptimal methods, it exhibits superior performance across multiple datasets.

**Weaknesses:**

（1）Figure 4: (a) is quite messy, and the curves for different network architectures cannot be clearly distinguished and compared. Please provide a more intuitive explanation and presentation method.
（2）Although RandomNAS-OGL has a slight advantage in test error rate in Table 2, PDARTS is superior in terms of model size and computational complexity.All things considered, I believe the latter is more superior than your model. Please find a better approach to optimize your model.
（3）The storage and computational overhead are issues you need to consider at the moment, as they will greatly limit you in real-world application scenarios.
（4）When comparing performance, many methods were not reprouced or their true performance metrics were not obtained. Therefore, I believe your comparison is lacking and not comprehensive.

**Questions:**

Please refer to the weaknesses.

**Details Of Ethics Concerns:**

I do not find any ethical problem in this paper.

---

> ### Author Response · Authors · 2023-11-20
> **Rebuttal by Authors**
>
> **Weakness1**: Thanks for your comments! It is true that Fig.4 is unable to see the relationships between different architectures, and we refer to [1] and find a new visualization method to show the changes of the validation accuracy of architectures and the impact between different architectures. New Figures and related descriptions have been correspondingly updated in our paper.
>
> [1] Shuaicheng Niu, Jiaxiang Wu, Yifan Zhang, Yong Guo, Peilin Zhao, Junzhou Huang, and Mingkui Tan. Disturbance-immune weight sharing for neural architecture search. Neural Networks, 144: 553–564, 2021.
>
> **Weakness2**: Thanks for your insightful and constructive advice. We agree with your opinion. The use of traditional RandomNAS as the baseline in this study is to verify the effectiveness of OGL in solving the multi-model forgetting issue. However, we think OGL will get better results with PDARTS for the following reasons. 1. PDARTS is an improvement over DARTS, especially on the storage to train large models. PDARTS divides the search process into three stages, and increases the depth of the network while reducing the types of operations. Therefore, PDARTS has lower search costs and smaller model size than others; 2. OGL projects the direction of weight update in baselines on the orthogonal direction of the gradient space, which has been proved in Lemma 1 that the orthogonal direction of weight update shows slightly influence on the performance of the previously trained architectures. In other words, the multi-model forgetting issue is largely alleviated.
>
> Based on the reasons above. We believe that applying OGL to PDARTS (PDARTS-OGL) can get better results. We will consider PDARTS-OGL in our future work and apply OGL to more other excellent methods.

---

> > ### Author Response · Authors · 2023-11-20
> > **continue to Rebuttal Part 2**
> >
> > **Weakness3**: Thanks for your constructive comments. It is true that our method needs to calculate the gradient vector of each operation and update the corresponding gradient space by storing the new gradient vectors, which may increase the storage and computational costs. In the following, we compared the methods in Table 1 to get more comprehensive understanding:
> >
> > |       Method        | Test Error (%) (C10) | Test Error (%) (C100) | Prarm. (M) | FLOPs (M) | Search Cost (GPU Days) |
> > | ------------------- | -------------------- | --------------------- | ---------- | --------- | ---------------------- |
> > | NAO-WS              | 3.53                 | -                     | 2.5        | -         | -                      |
> > | ENAS                | 3.54                 | 19.43                 | 4.6        | -         | 0.45                   |
> > | SNAS                | 2.85 $\pm$ 0.02      | 20.09                 | 2.8        | 422       | 1.5                    |
> > | PARSEC              | 2.86 $\pm$ 0.06      | -                     | 3.6        | 485       | 0.6                    |
> > | BayesNAS            | 2.81 $\pm$ 0.04      | -                     | 3.4        | -         | 0.2                    |
> > | RENAS               | 2.88 $\pm$ 0.02      | -                     | 3.5        | -         | 6                      |
> > | MdeNAS              | 2.87                 | 17.61                 | 3.78       | 599       | 0.16                   |
> > | DSO-NAS             | 2.87 $\pm$ 0.07      | -                     | 3          | -         | 1                      |
> > | Random Baseline     | 3.29 $\pm$ 0.15      | -                     | 3.2        | -         | 4                      |
> > | DARTS(1st)          | 2.94                 | -                     | 2.9        | 501       | 1.5                    |
> > | DARTS(2nd)          | 2.76 $\pm$ 0.09      | 17.57                 | 3.4        | 528       | 4                      |
> > | WPL                 | 3.81                 | -                     | -          | -         | -                      |
> > | DI-RandomNAS+cutout | 2.87 $\pm$ 0.04      | 17.71                 | 3.6        | -         | 1.5                    |
> > | RandomNAS-NSAS      | 2.64                 | 17.56                 | 3.08       | 489       | 0.7                    |
> > | GDAS-NSAS           | 2.73                 | 18.02                 | 3.54       | 528       | 0.4                    |
> > | GDAS                | 2.93                 | 18.38                 | 3.4        | 519       | 0.21                   |
> > | **GDAS-OGL**        | 2.83                 | 17.75                 | 3.14       | 528       | 0.3                    |
> > | RandomNAS           | 2.85 $\pm$ 0.08      | 17.63                 | 4.3        | 612       | 2.7                    |
> > | **RandomNAS-OGL**   | 2.63 $\pm$ 0.02      | 17.54                 | 3.52       | 503       | 0.5                    |
> >
> > From the above table,  the FLOPs and search cost indicate the computional costs.  From the table, OGL shows a slighly higher FLOPs but relatively lower search costs (GPU days) in comparison with other methods. OGL aims to deal with multi-model forgetting issue and have to calculate the gradient vector of each operation and update the corresponding gradient space. Hence, the improvement of the accuracy (Test error) achieved by OGL is on the sacrifice of the complexity of the proposed method, leading to a higher FLOPs. Notably, the reasons for fewer GPU days (e.g., 0.3 and 0.5 GPU days on RandomNAS-OGL and GDAS-OGL, respectively) of OGL is because the improvement of the accuracy can speed up the model convergence rate. From the results, OGL is relatively efficient in supernet training.
> >
> > For the storage issue, although PCA has reduced the dimension of the gradient space, OGL still needs to store a set of base vectors. Hence, we will develop more efficent calculation of orthogonal direction with fewer storage, with the consideration of using iterative methods instead of directly storing them.
> >
> > The above issues are explained in section 5 in the manuscript.

---

> > > ### Author Response · Authors · 2023-11-20
> > > **continue to Rebuttal Part 3**
> > >
> > > **Weakness4**: Thanks for your constructive suggestions. We have made a more comprehensive experiments as follows:
> > >
> > > |       Method        | Test Error (%) (C10) | Test Error (%) (C100) | Prarm. (M) | Search Cost (GPU Days) |
> > > | ------------------- | -------------------- | --------------------- | ---------- | ---------------------- |
> > > | ENAS                | 3.54                 | 19.43                 | 4.6        | 0.45                   |
> > > | SNAS                | 2.85 $\pm$ 0.02      | 20.09                 | 2.8        | 1.5                    |
> > > | MdeNAS              | 2.87                 | 17.61                 | 3.78       | 0.16                   |
> > > | DARTS(2nd)          | 2.76 $\pm$ 0.09      | 17.57                 | 3.4        | 4                      |
> > > | MetaQNN [1]         | 6.92                 | 27.14                 | 11.2       | >80                    |
> > > | NASNet-A [2]        | 3.41                 | 19.7                  | 3.3        | 2000                   |
> > > | NASNet-A+CutOut [2] | 2.65                 | 17.81                 | 3.3        | 2000                   |
> > > | SMASH [3]           | 4.03                 | 20.6                  | 16         | -                      |
> > > | GDAS + cutout [4]   | 3.93                 | 18.38                 | 3.4        | 0.84                   |
> > > | ENAS + cutout [5]   | 2.89                 | 18.91                 | 4.6        | 0.5                    |
> > > | WPL                 | 3.81                 | -                     | -          | -                      |
> > > | DI-RandomNAS+cutout | 2.87 $\pm$ 0.04      | 17.71                 | 3.6        | 1.5                    |
> > > | RandomNAS-NSAS      | 2.64                 | 17.56                 | 3.08       | 0.7                    |
> > > | GDAS-NSAS           | 2.73                 | 18.02                 | 3.54       | 0.4                    |
> > > | GDAS                | 2.93                 | 18.38                 | 3.4        | 0.21                   |
> > > | **GDAS-OGL**        | 2.83                 | 17.75                 | 3.14       | 0.3                    |
> > > | RandomNAS           | 2.85 $\pm$ 0.08      | 17.63                 | 4.3        | 2.7                    |
> > > | **RandomNAS-OGL**   | **2.63 $\pm$ 0.02**  | **17.54**             | 3.52       | 0.5                    |
> > >
> > > In order to evaluate the performance of the proposed OGL in handling the multi-model forgetting issue, we compared OGL with WPL, DI-RandomNAS, RandomNAS-NSAS and GDAS-NSAS, which are the existing methods in solving the multi-model forgetting issue. And the experimental results demonstrate that our OGL performs better in solving the multi-model forgetting. Specifically, RandomNAS-OGL and GDAS-OGL outperforms WPL with a test error improvement of 1.18% and 0.98% on CIFAR-10, respectively. Also, RandomNAS-OGL performs better than RandomNAS-NSAS with 0.01% test error improvement on CIFAR-10 and 0.02% test error improvement on CIFAR-100. Furthermore, GDAS-OGL performs better than GDAS-NSAS on CIFAR-100 with smaller test error (0.27% improvement) and less parameters (0.4M improvement). In addition, RandomNAS-OGL outperforms DI-RandmNAS with a test error improvement of 0.24% and 0.17% on CIFAR-10 and CIFAR-100, respectively.
> > >
> > > In addition, we also compared OGL with ten state-of-the-art one-shot NAS methods without designing techniques to handle the multi-model forgetting issues. In Table 1, we have also provided extra six state-of-the-art one-shot NAS methods to get a comprehensive comparison. Specifically, RandomNAS-OGL gets 2.63% test error on CIFAR-10 and 17.54% test error on CIFAR-100, and GDAS-OGL achieves 2.83% on CIFAR-10 and 17.75% on CIFAR-100, which are better than most of compared methods. The results show that the methods especially OGL with the techniques show better performance than other NAS methods without the techniques.
> > >
> > > [1] Bowen Baker, Otkrist Gupta, Nikhil Naik, and Ramesh Raskar. Designing neural network architectures using reinforcement learning. In International Conference on Learning Representations (ICLR), 2017.
> > >
> > > [2] Barret Zoph, Vijay Vasudevan, Jonathon Shlens, and Quoc V. Le. Learning transferable architectures for scalable image recognition. In IEEE Conference on Computer Vision and Pattern Recognition (CVPR), pages 8697–8710, 2018.
> > >
> > > [3] Andrew Brock, Theodore Lim, James M Ritchie, and Nick Weston. Smash: One-shot model archi-tecture search through hypernetworks. arXiv preprint arXiv:1708.05344, 2017.
> > >
> > > [4] Xuanyi Dong and Yi Yang. Searching for a robust neural architecture in four gpu hours. In Computer Vision and Pattern Recognition (CVPR), pp. 1761–1770, 2019b.
> > >
> > > [5] Xuanyi Dong and Yi Yang. Searching for a robust neural architecture in four gpu hours. In Computer Vision and Pattern Recognition (CVPR), pp. 1761–1770, 2019b.

---

> ### Author Response · Authors · 2023-11-23
> **Looking forward to your feedback**
>
> Dear Reviewer,
>
> We would like to express our sincere gratitude for taking the time to review our paper and providing valuable feedback. We appreciate your expertise and insights, which are significant in enhancing the quality of our work.
>
> We are eagerly awaiting your response to our rebuttal, as your comments and suggestions are crucial to the further development of our research. We understand that your time is precious, and we respect your commitment to this process.
>
> In our rebuttal, we have thoroughly addressed the issues raised and provided comprehensive discussions and revisions. If there is any additional information you require or further clarification needed, please do not hesitate to reach out to us. We are more than willing to explain in any way possible.
>
> Once again, we appreciate your dedication and effort in reviewing our paper. Your prompt attention to our rebuttal would be highly appreciated, as it greatly contributes to the timely progress of our research.
>
> Thank you for your consideration, and we look forward to receiving your valuable feedback.
>
> Sincerely,
>
> The authors of Submission6907

---

### Official Review · Reviewer_iP4N · 2023-11-01

**Soundness:** 2 fair
**Presentation:** 2 fair
**Contribution:** 2 fair
**Rating:** 5
**Confidence:** 3

**Summary:**

The work focus on solving the problem of multi-model forgetting in neural architecture search (NAS) . To address this problem, the authors propose an Orthogonal Gradient Learning (OGL) for one-shot NAS-guided supernet training. This method updates the weights of the overlapping structures of the current architecture in directions orthogonal to the gradient space of these structures in all previously trained architectures.

The authors provide experimental evidence supporting the effectiveness of the proposed paradigm in mitigating multi-model forgetting.

**Strengths:**

the authors propose the Orthogonal Gradient Learning (OGL) . This method updates the weights of the overlapping structures of the current architecture in directions orthogonal to the gradient space of these structures in all previously trained architectures.

The authors provide experimental evidence supporting the effectiveness of the proposed paradigm in mitigating multi-model forgetting.

**Weaknesses:**

The proposed orthogonal gradient learning (OGL) guided supernet training method may be sensitive to hyperparameters. The paper should conduct a more detailed analysis of the impact of hyperparameters on the robustness of the method.

This paper mentions the theoretical support for the proposed approach, but the assumptions made in these theoretical proofs and their relevance to actual NAS scenarios should be detailed.

**Questions:**

This paper aims to reduce the computational budget in NAS; it should provide a more complete analysis of the computational cost introduced by the OGL approach, as this can be an issue in resource-constrained environments.

---

> ### Author Response · Authors · 2023-11-20
> **Rebuttal by Authors**
>
> **Weakness1**: Many thanks for your comments! OGL is involved in a hyperparameter $d$ in $G_{dim}\in \mathbb{R}^{h \times d}$, which is the column dimension of the gradient space, i.e., the number of the gradient vectors. However, the parameter $d$ is self-adjusted with the optimization process. Specifically, it is known that a space can be spanned by the maximally linear independent set, and PCA used in this study is to find the independent set of the gradient space. Once the independent set is found, $d$ here is determined since it is set to the number of vectors in the maximally linear independent set. Accordingly, we did not perform the parameter analysis of the proposed method.
>
> **Weakness2**:  Many thanks for your valuable feedback on the assumptions made in the theoretical proofs.
>
> In the proof of lemma 1, there is a remainder term (i.e.,$R_1(w)$) of Taylor expansion of the loss function, namely, the infinitesimal of the first order.  In practice, we ignored this term for theoretical proof and drew a conclusion that the update of weights along the orthogonal direction of the gradient of this operation will slightly change the accurancy of the trained architecture. However, we must admit that ignoring $R_1(w)$ will impact the accuracy of the trained architecture to some extent, but the impact is tiny. Accordingly, we have emphasized the conclusion in the manuscript that the OGL guided training paradigm enables the training of the current architecture to largely eliminate the impact to the performance of all previously trained architectures.

---

> > ### Author Response · Authors · 2023-11-20
> > **continue to Rebuttal Part 2**
> >
> > **Q1**: This paper aims to reduce the computational budget in NAS; it should provide a more complete analysis of the computational cost introduced by the OGL approach, as this can be an issue in resource-constrained environments.
> >
> > **A1**: Thanks for your comments. Please find the analysis of the computational costs of the proposed OGL as follows:
> >
> > |       Method        | Test Error (%) (C10) | Test Error (%) (C100) | Prarm. (M) | FLOPs (M) | Search Cost (GPU Days) |
> > | ------------------- | -------------------- | --------------------- | ---------- | --------- | ---------------------- |
> > | NAO-WS              | 3.53                 | -                     | 2.5        | -         | -                      |
> > | ENAS                | 3.54                 | 19.43                 | 4.6        | -         | 0.45                   |
> > | SNAS                | 2.85 $\pm$ 0.02      | 20.09                 | 2.8        | 422       | 1.5                    |
> > | PARSEC              | 2.86 $\pm$ 0.06      | -                     | 3.6        | 485       | 0.6                    |
> > | BayesNAS            | 2.81 $\pm$ 0.04      | -                     | 3.4        | -         | 0.2                    |
> > | RENAS               | 2.88 $\pm$ 0.02      | -                     | 3.5        | -         | 6                      |
> > | MdeNAS              | 2.87                 | 17.61                 | 3.78       | 599       | 0.16                   |
> > | DSO-NAS             | 2.87 $\pm$ 0.07      | -                     | 3          | -         | 1                      |
> > | Random Baseline     | 3.29 $\pm$ 0.15      | -                     | 3.2        | -         | 4                      |
> > | DARTS(1st)          | 2.94                 | -                     | 2.9        | 501       | 1.5                    |
> > | DARTS(2nd)          | 2.76 $\pm$ 0.09      | 17.57                 | 3.4        | 528       | 4                      |
> > | WPL                 | 3.81                 | -                     | -          | -         | -                      |
> > | DI-RandomNAS+cutout | 2.87 $\pm$ 0.04      | 17.71                 | 3.6        | -         | 1.5                    |
> > | RandomNAS-NSAS      | 2.64                 | 17.56                 | 3.08       | 489       | 0.7                    |
> > | GDAS-NSAS           | 2.73                 | 18.02                 | 3.54       | 528       | 0.4                    |
> > | GDAS                | 2.93                 | 18.38                 | 3.4        | 519       | 0.21                   |
> > | **GDAS-OGL**        | 2.83                 | 17.75                 | 3.14       | 528       | 0.3                    |
> > | RandomNAS           | 2.85 $\pm$ 0.08      | 17.63                 | 4.3        | 612       | 2.7                    |
> > | **RandomNAS-OGL**   | 2.63 $\pm$ 0.02      | 17.54                 | 3.52       | 503       | 0.5                    |
> >
> > We have compared the computational costs of all methods in Table 1, where the FLOPs and search cost indicate the computional costs. From the table, OGL shows a slighly higher FLOPs but relatively lower search costs (GPU days) in comparison with other methods. OGL aims to deal with multi-model forgetting issue and have to calculate the gradient vector of each operation and update the corresponding gradient space. Hence,  the improvement of the accuracy achieved by OGL is on the sacrifice of the complextity of the proposed method, leading to a higher FLOPs. Notably, the reasons for fewer GPU days of OGL is because the improvement of the accuracy can speed up the model convergence rate.

---

> ### Author Response · Authors · 2023-11-23
> **Looking forward to your feedback**
>
> Dear Reviewer,
>
> We would like to express our sincere gratitude for taking the time to review our paper and providing valuable feedback. We appreciate your expertise and insights, which are significant in enhancing the quality of our work.
>
> We are eagerly awaiting your response to our rebuttal, as your comments and suggestions are crucial to the further development of our research. We understand that your time is precious, and we respect your commitment to this process.
>
> In our rebuttal, we have thoroughly addressed the issues raised and provided comprehensive discussions and revisions. If there is any additional information you require or further clarification needed, please do not hesitate to reach out to us. We are more than willing to explain in any way possible.
>
> Once again, we appreciate your dedication and effort in reviewing our paper. Your prompt attention to our rebuttal would be highly appreciated, as it greatly contributes to the timely progress of our research.
>
> Thank you for your consideration, and we look forward to receiving your valuable feedback.
>
> Sincerely,
>
> The authors of Submission6907

---

### Official Review · Reviewer_74aa · 2023-11-04

**Soundness:** 2 fair
**Presentation:** 3 good
**Contribution:** 2 fair
**Rating:** 5
**Confidence:** 5

**Summary:**

This paper proposes a new method called Orthogonal Gradient Learning (OGL) to overcome multi-model forgetting in one-shot NAS. It updates weights of overlapped structures in the orthogonal direction to the gradient space of previously trained architectures. This avoids overwriting well-trained models while training new architectures sequentially. A PCA-based projection is used to find orthogonal directions without storing all past gradient vectors. OGL is integrated into RandomNAS and GDAS one-shot NAS baselines. Experiments show OGL reduces forgetting, leading to better final architectures and stronger supernet predictive ability.

**Strengths:**

**Strengths**:

- Original idea of using orthogonal gradient updates to avoid catastrophic forgetting in NAS.

- Technically sound approach grounded in theory with clear algorithm design and experimental methodology.

- Strong empirical results demonstrating reduced forgetting and improved search performance compared to baselines.

- The PCA-based projection to compute orthogonal gradients is creative and helps address a key limitation.

- OGL seems widely applicable to enhance different one-shot NAS methods as shown by results on two baselines.

**Weaknesses:**

**Weaknesses**:

- Theoretical analysis is limited, more formal convergence guarantees could strengthen the approach.

- Certain details like schedule for gradient space updates are unclear. Sensitivity to hyper-parameters not fully studied.

- Experiments focus on small CNN search spaces, evaluating on larger spaces like transformers could be useful.

- Qualitative analysis into why and how OGL architectures differ from baseline NAS would provide more insight.

- Extending OGL to other architecture search domains like hyperparameter optimization could further demonstrate generality.

**Questions:**

see Weaknesses

---

> ### Author Response · Authors · 2023-11-20
> **Rebuttal by Authors**
>
> **Weakness1**: Thanks for your constructive comments. We provide with the convergence analysis of the update of the weights as follows:
>
> ***Theorem* 1 :** Given a $l$-smooth and convex loss function $L(w)$, $w^*$ and $w_0$ are the optimal and initial weights of $L(w)$, respectively. If we let the learning rate $\eta = 1/l$, then we have:
>
> $$
> L(w_t) - L(w^*) \leq \frac{2l}{t} \left\lVert w_0 - w^* \right\rVert_F^2,
> $$
> where $w_t$ is the weights of architecture after $t$-th training.
>
> According to *Theorem*  1, the OGL has a convergence rate of $O(1/t)$. The main proof of Theorem 1 is as follows:
>
> **Proof.** The update of the weights of architecture $w$ can be represented as follows:
>
> $$
> q(w_t) = \arg\min_{w \in Q} (L(w_t) + \langle L^{'}(w_t), w - w_t \rangle + \frac{\beta}{2}\left\lVert w_t - w \right\rVert_F^2)
> $$
> $$
> w_{t+1} = w_t - \eta\beta(w_t - q(w_t)),
> $$
> where $\eta$ and $\beta$ are two hyperparameters.
>
> Let $Q$ is a closed convex set, $w^+ \in Q$ and $\beta \geq l$. We denote $Q_w = q(w^+)$ and $g_Q = g_Q(w^+) = \beta(w^+ - q(w^+))$, then we have:
>
> $$
> \langle L^{'}(w^+) - g_Q, w - Q_w \rangle = \langle L^{'}(w^+), w - Q_w \rangle - \langle g_Q, w - Q_w \rangle \geq 0.
> $$
>
> And combined with the property of convex function, we have:
>
> $$
> L(w) \geq L(Q_w) + \frac{1}{2\beta}\left\lVert g_Q \right\rVert_F^2 + \langle g_Q, w - w^+ \rangle.
> $$
>
> Let $\beta = l$, $w = w^+ = w_t$, we have:
>
> $$
> L(w_t) \geq L(w_{t+1}) + \frac{1}{2l}\left\lVert g_Q(w_t) \right\rVert_F^2,
> $$
> where $g_Q(w_t) = \beta(w_t - q(w_t))$.
>
> Let $\beta = l$, $w = w^*$, $w^+ = w_t$, we have:
>
> $$
> L(w^*) \geq L(w^*) + \frac{1}{2l}\left\lVert g_Q(w_t) \right\rVert_F^2 - \langle g_Q(w_t), w^* - w^t \rangle.
> $$
>
> We denote $r_t = \left\lVert w_t - w^* \right\rVert_F$ and $g_{Q,t} = g_Q(w_t)$, then if $\eta \leq \frac{1}{\beta}$ we have:
>
> $$
> r_{t+1}^2 \leq r_t^2 + \eta (\eta - \frac{1}{\beta}) \left\lVert g_{Q,t} \right\rVert_F^2 \leq r_t^2 \leq \ldots \leq r_0^2.
> $$
>
> We denote $\Delta_t = L(w_t) - L(w^*)$, and $\Delta_{t+1} \leq \Delta_t$, we have:
>
> $$
> \frac{1}{\Delta_{t+1}} \geq \frac{1}{\Delta_{t}} + \frac{1}{2l} \frac{1}{r_0^2} \frac{\Delta_{t}}{\Delta_{t+1}} \geq \frac{1}{\Delta_t} + \frac{1}{2l} \frac{1}{r_0^2} \geq \ldots \geq \frac{1}{\Delta_0} + \frac{1}{2l} \frac{1}{r_0^2}.
> $$
>
> Then we have:
>
> $$
> \Delta_t = \frac{2l (L(w_0) - L(w^*)) \left\lVert w_0 - w^* \right\rVert_F^2}{2l \left\lVert w_0 - w^* \right\rVert_F^2 + t (L(w_0) - L(w^*))}.
> $$
>
> Based on the property of $l$-smooth function, we have:
>
> $$
> L(w_0) \leq L(w^*) + \langle L^{'}(w^*), w_0 - w^* \rangle + \frac{l}{2} \left\lVert w_0 - w^* \right\rVert_F^2.
> $$
>
> Based on the inequation above, we have:
>
> $$
> L(w_t) - L(w^*) \leq \frac{2l \left\lVert w_0 - w^* \right\rVert_F^2}{2l \frac{\left\lVert w_0 - w^* \right\rVert_F^2}{L(w_0) - L(w^*)} + t} \leq \frac{2l}{t} \left\lVert w_0 - w^* \right\rVert_F^2.
> $$
>
> Thus the Theorem 1 is proven.
>
> The details of the proof of Theorem 1 is provided in the section 1.2 of the Supplementary material.
>
>
> **Weakness2**: Sorry for the unclear statement and we detailed the update of the gradient space as follows:
>   1. Initialization of the gradient spaces: We firstly design a gradient space $G$ for each operation to save the gradient vectors of corresponding operation, then initialize each gradient space as the identity matrix $I$ (i.e.,  $G = I$);
>   2. Training of architectures: During the process of supernet training, an architecture is sampled in each supernet training epoch, and then the weights of the architecture is updated through OGL;
>   3. Calculation of the gradient vectors: We calculate the gradient of the weights for each operation in the supernet after the update of the architecture, and then obtain the gradient vectors of each operation (i.e., $g_1, g_2, ..., g_n$ where $n$ is the number of gradient vectors);
>   4. Concatenation of gradient vectors:  We concatenate the gradient vectors $g_1, g_2, ..., g_n$ to the gradient space $G$ (i.e., $G\leftarrow [G, g_1, g_2, ..., g_n]$);
>   5. Dimension reduction of gradient space through PCA: We extract a set of base vectors of each gradient space by PCA (i.e., $G\in \mathbb{R}^{h \times n} \rightarrow G_{dim}\in \mathbb{R}^{h \times d}$, where $h$ and $d$ are the dimension of the gradient space and the number of base vectors, respectively).
>
> About the hyperparameters. OGL is involved in a hyperparameter $d$ in $G_{dim}\in \mathbb{R}^{h \times d}$, which is the column dimension of the gradient space, i.e., the number of the gradient vectors. However, the parameter $d$ is self-adjusted with the optimization process. Specifically, it is known that a space can be spanned by the maximally linear independent set, and PCA used in this study is to find the independent set of the gradient space. Once the independent set is found, $d$ here is determined since it is set to the number of vectors in the maximally linear independent set. Accordingly, we did not perform the parameter analysis of the proposed method.

---

> ### Author Response · Authors · 2023-11-20
> **continue to Rebuttal Part 2**
>
> **Weakness3**: Yes. We totally agree with your opinion and thanks for your comments. Theoretically, our method is more competitive by using a larger search space than a smaller one. Specifically,  more candidate architectures will be sampled from a larger search space to train the supernet to get high performance, leading to more coupling candidate architectures. Therefore,  the multi-model forgetting issue is more likely to occur. Hence, our method is more competitive by alleviating the issue by using a larger search space than a smaller one.
>
> Here, we in this study did not use a large search space and aim to get a fair comparison with the up-to-date methods in handling the multi-model forgetting issue, where most of them use small search space. However, we will experimentally investigate the performance of our method by using a larger search space to evaluate the comprehensive performance of our method.
>
> **Weakness4**: Many thanks and the differences between OGL and baseline NAS are as follows:
>
> The difference results from the multi-model forgetting issue in the baseline NAS. The key of the baseline NAS and other one-shot NAS methods is weight sharing, where the weights of all candidate architectures directly inherit from a supernet without training from scratch. However, it may introduce multi-model forgetting. During the training of a supernet, a number of architectures are sequentially sampled from the supernet and trained independently. Once the architectures have partially overlapped structures, the weights of these overlapped structures of the previously well-trained architecture will be overwritten by the weights of the newly sampled architectures. In this way, the performance of the previously well-trained architecture may be decreased. Therefore, we proposed a strategy called OGL to solve the multi-model forgetting exists in baseline NAS and other one-shot NAS methods.
>
> The difference between OGL and baseline NAS is on the weight update. OGL firstly detects the structures of current architecture whether they are ever sampled or not. If not, then the weights of the architecture are updated with back-propagation algorithm (e.g., SGD) and a pre-constructed gradient space is updated by the gradient direction of the structure. If yes, then the weights of the overlapped structures of the current architecture are updated in the orthogonal direction to the gradient space of all previously trained architectures. Following the OGL guided training paradigm, the training of the current architecture will largely eliminate the influence to the performance of all previously trained architectures, which has been proved in Lemma 1. In other words, the multi-model forgetting issue is largely alleviated.
>
> **Weakness5**: Many thanks for your insightful comments, and we will explore the generality of our method on other architecture search domains. It will be great interesting to extend OGL to other architecture search if the previously-trained and newly-sampled architectures have overlapped structures during the training.

---

> ### Author Response · Authors · 2023-11-23
> **Looking forward to your feedback**
>
> Dear Reviewer,
>
> We would like to express our sincere gratitude for taking the time to review our paper and providing valuable feedback. We appreciate your expertise and insights, which are significant in enhancing the quality of our work.
>
> We are eagerly awaiting your response to our rebuttal, as your comments and suggestions are crucial to the further development of our research. We understand that your time is precious, and we respect your commitment to this process.
>
> In our rebuttal, we have thoroughly addressed the issues raised and provided comprehensive discussions and revisions. If there is any additional information you require or further clarification needed, please do not hesitate to reach out to us. We are more than willing to explain in any way possible.
>
> Once again, we appreciate your dedication and effort in reviewing our paper. Your prompt attention to our rebuttal would be highly appreciated, as it greatly contributes to the timely progress of our research.
>
> Thank you for your consideration, and we look forward to receiving your valuable feedback.
>
> Sincerely,
>
> The authors of Submission6907

---

### Meta-Review · Area_Chair_2gSP · 2023-12-13

**Metareview:**

This submission tackles the problem of multi-model forgetting during supernet training for one-shot NAS. The weights of a previously trained architecture's operation can be overwritten by the updates of a newly sampled architecture. The authors propose an orthogonal gradient learning guided supernet training - update weights in a direction that is orthogonal to the gradient space of all previously trained architectures.
The reviewers appreciated the novel idea proposed for one-shot NAS, but also pointed out a number of issues with the manuscript in its current form. Multiple reviewers complained that the theoretical analysis provided in the submission was insufficient, and that crucial details such as storage and computational overhead, and insightful experiments were missing.
After discussion amongst the ACs, it was agreed that the manuscript needed significant updates and was lacking in its current form.

**Justification For Why Not Higher Score:**

Multiple Reviewers raised issues regarding insufficient theoretical analysis and missing details. The submission needs a significant overhaul to address all the issues.

**Justification For Why Not Lower Score:**

N/A

---

### Decision · Program_Chairs · 2024-01-16

Reject